# Possible Mechanisms of Eosinophil Accumulation in Eosinophilic Pneumonia

**DOI:** 10.3390/biom10040638

**Published:** 2020-04-21

**Authors:** Kazuyuki Nakagome, Makoto Nagata

**Affiliations:** 1Department of Respiratory Medicine, Saitama Medical University, Saitama 350-0495, Japan; favre4mn@saitama-med.ac.jp; 2Allergy Center, Saitama Medical University, Saitama 350-0495, Japan

**Keywords:** chemokines, cytokines, eosinophilic pneumonia, eosinophils, pneumonia

## Abstract

Eosinophilic pneumonia (EP), including acute EP and chronic EP, is characterized by the massive pulmonary infiltration of eosinophils into the lung. However, the mechanisms underlying the selective accumulation of eosinophils in EP have not yet been fully elucidated. We reported that bronchoalveolar lavage fluid (BALF) from EP patients induced the transmigration of eosinophils across endothelial cells in vitro. The concentrations of eotaxin-2 (CCL24) and monocyte chemotactic protein (MCP)-4 (CCL13), which are CC chemokine receptor (CCR) 3 ligands, were elevated in the BALF of EP patients, and anti-CCR3 monoclonal antibody inhibited the eosinophil transmigration induced by the BALF of EP patients. The concentration of macrophage inflammatory protein 1β (CCL4), a CCR5 ligand that induces eosinophil migration, was increased in the BALF of EP patients. Furthermore, the concentration of interleukin (IL) 5 was increased in the BALF of EP patients, and it has been reported that anti-IL-5 antibody treatment resulted in remission and the reduction of glucocorticoid use in some cases of chronic EP. The concentrations of lipid mediators, such as leukotriene (LT) B_4_, damage-associated molecular pattern molecules (DAMPs), such as uric acid, or extracellular matrix proteins, such as periostin, were also increased in the BALF of EP patients. These findings suggest that chemokines, such as CCR3/CCR5 ligands, cytokines, such as IL-5, lipid mediators, such as LTB_4_, DAMPs, and extracellular matrix proteins may play roles in the accumulation or activation of eosinophils in EP.

## 1. Introduction

Eosinophilic pneumonia (EP) is characterized by the massive pulmonary infiltration of eosinophils into the lung [1,2,3,4]. EP encompasses a variety of lung diseases with a heterogeneous background, and its prevalence has not been fully clarified, likely due to the heterogeneity, at least in part. Eosinophilic lung diseases are classified as EP of undetermined cause, EP of determined cause, and a miscellaneous group of lung diseases [1,2]. EP of undetermined causes include idiopathic EP, such as acute eosinophilic pneumonia (AEP) [3] and chronic eosinophilic pneumonia (CEP) [4], and EP associated with systemic diseases, such as eosinophilic granulomatosis with polyangiitis (EGPA) and hypereosinophilic syndrome. EP of determined causes include EP secondary to parasitic or fungal infection, drug-induced reactions, and allergic bronchopulmonary micosis (ABPM). The miscellaneous group of lung diseases includes organizing pneumonia and idiopathic interstitial pneumonia. Certain drugs, chemical fumes, molds, and cigarette smoke can induce EP [1,2]. Molds can induce EP through fungal infection, or an allergic reaction such as ABPM. However, the mechanisms underlying the accumulation of eosinophils in EP have not yet been fully established. In this review, possible mechanisms of eosinophil accumulation in the airway of EP patients are discussed. The aim of this review is to better understand the mechanisms of eosinophil accumulation and activation in EP.

## 2. AEP and CEP

AEP is characterized by acute febrile illness with diffuse pulmonary infiltrates, severe hypoxemia, and increased eosinophils in bronchoalveolar lavage fluid (BALF) [3]. AEP is diagnosed based on the following: acute onset of respiratory failure, diffuse pulmonary infiltrates on chest roentgenogram, and increased numbers of eosinophils in BALF (more than 25% of total cells) [3]. Although the mechanisms of AEP have not yet been fully established, inhaled agents, such as cigarette smoke or chemical agents, are known causes of AEP [1,2,5,6,7,8,9]. For example, a relationship was observed between the recent onset of cigarette smoking and the development of AEP [5,6,7]. Even short-term passive smoking can induce AEP [7]. The collapse of the World Trade Center towers [8] and the desert of the Middle East [9] have also been reported to induce AEP.

AEP is such a rare disease that its prevalence has not been fully elucidated. In a study of United States military personnel in the Middle East, the estimated prevalence of AEP was 9.1 cases per 100,000 person-years [9], although their inhalational exposure differs from the general condition. Most AEP patients are around 20 years of age, and males and current smokers are more predominant; however, the AEP patients do not have allergic diseases, such as asthma [1,2]. Dyspnea, cough, and fever that all develop within several days are found in most patients [1,2]. Blood eosinophil counts are normal in most AEP patients at the time of onset (or the time of admission), then transiently decrease by case, but they subsequently increase. The typical findings of computed tomography (CT) in AEP are shown in Figure 1. Ground-glass opacity/airspace consolidation and interlobular septal thickening are representative CT findings in AEP [1,2]. Eosinophilia in BALF is important for the diagnosis of AEP. Systemic corticosteroids rapidly improve AEP within several days, and spontaneous resolution can be expected in mild cases. AEP does not usually recur [1,2].

CEP is a slowly progressive eosinophilic lung disease with an unknown cause [4]. The characteristics of CEP include: dense and multiple foci of consolidation in peripheral lung fields on chest roentgenogram, increased percentage of eosinophils in BALF, absence of other possible causes, and a favorable response to systemic corticosteroid therapy [4]. CEP is also a rare disease. The prevalence of CEP among interstitial lung diseases was reported to be 0% to 2.7% in an interstitial lung disease registry in Europe and in the United States [10]. Sveinsson et al. reported that the annual incidence of CEP was estimated to be 23 cases per 100,000 of the population in Iceland [11]. Most CEP patients are around 30 to 50 years of age, and there is a female predominance [1,2]. Some cases of CEP are related to bronchial asthma: some have pre-existing asthma and others develop asthma after the onset of CEP [1,2]. There are no symptoms that are specific to CEP, although a cough and shortness of breath are common. Blood eosinophil counts are elevated in most CEP patients [1,2]. Typical CT findings in CEP are shown in Figure 1. Bilateral or unilateral airspace consolidation, predominantly in the peripheral region (photographic negative pattern), is a representative CT finding of CEP [1,2]. Eosinophilia in BALF is also important for the diagnosis of CEP. Systemic corticosteroids usually also improve CEP. However, in contrast to AEP, patients with CEP often relapse at some point [1,2].

The levels of various cytokines and chemokines, including type 2 cytokines, are increased in the BALF of patients with EP [12,13,14,15,16,17,18], and these cytokines and chemokines are thought to play roles in eosinophil accumulation and activation. The concentrations of the cytokines/chemokines in BALF are higher in AEP than in CEP, although the eosinophil count is similar in both [12,13,14,15,16,17,18,19]. In fact, the concentrations of interleukin (IL) 5 and IL-13 in BALF are higher in AEP than in CEP [12,13,14,15,16,17,19]. Furthermore, compared to CEP, AEP involves the higher airway production of chemokines, such as eotaxin, for the activation of eosinophils, and epithelial cell-derived cytokines, such as IL-33, for the induction of the type 2 response [15,16,17,18].

## 3. Mechanisms for the Development of Eosinophilic Airway Inflammation

Eosinophils accumulate at sites of allergic inflammation and play important roles in the pathogenesis of diseases, including EP, through the release of a variety of mediators, including specific granule proteins, such as major basic protein, cysteinyl leukotrienes (cysLTs), radical oxygen species, and cytokines [20]. Among them, IL-5 is a well-recognized cytokine associated with eosinophilic inflammation. The in vitro effects of IL-5 on eosinophils include the priming of the eosinophils and the enhancement of their survival [21]. Recently, the roles of IL-5 in the pathogenesis of eosinophilic diseases, such as eosinophil-dominant severe asthma and EGPA, have been established. In severe asthmatics with persistent blood or sputum eosinophilia, anti-IL-5 monoclonal antibody (mAb) treatment reduced both the blood eosinophil count and acute exacerbations [22,23,24]. In EGPA, anti-IL-5 mAb treatment resulted in a higher proportion of remission, thus allowing for the reduced use of glucocorticoid [25]. The concentration of IL-5 is increased in the BALF of EP patients, as described above. In this context, several case reports have suggested a positive effect for mepolizumab (anti-IL-5 mAb) on remission and the reduction of glucocorticoid use in CEP [26,27].

Eosinophils must adhere to and migrate across vascular endothelial cells to accumulate in airways [28,29]. These processes are mainly regulated by cytokines, especially IL-4/IL-13, or chemokines from a variety of cells, such as T helper (Th) 2 cells or group 2 innate lymphoid cells. The interaction between eosinophils and endothelial cells via the α4 integrins/vascular cell adhesion molecule (VCAM)-1 pathway is an important step for selective eosinophil recruitment [28,29]. IL-4 and IL-13 induce VCAM-1 expression on endothelial cells, and blood eosinophils spontaneously adhere to VCAM-1 [30,31]. This interaction between eosinophils and VCAM-1 induces superoxide anion (O_2_^−^) generation and the degranulation of eosinophils, and can be considered the initial step of eosinophil activation [28,29,30,31,32].

Eosinophil chemoattractants can induce the effective migration of eosinophils after adhesive interaction with endothelial adhesion molecules, such as VCAM-1 [28,29]. CC chemokines, especially CC chemokine receptor (CCR) 3 ligands, such as those regulated upon activation, normal T-expressed and secreted (RANTES; CCL5), eotaxin (CCL11), eotaxin-2 (CCL24), monocyte chemotactic protein (MCP)-3 (CCL7), and MCP-4 (CCL13), all effectively induce eosinophil transmigration across endothelial cells that express VCAM-1 [33]. Several studies have confirmed that CCR3 ligands are upregulated in the airways of bronchial asthma or EP patients [34,35,36,37]. Furthermore, the role of CCR5 and its ligands in the development of eosinophilic inflammation has been reported [38]. The concentration of macrophage inflammatory protein (MIP) 1β (CCL4), a CCR5 ligand, is increased in the BALF of EP patients [38].

Eosinophils can be activated and degranulated by cytokine/eosinophil growth factors, especially IL-5, IL-3, or granulocyte-macrophage colony-stimulating factor (GM-CSF), after migration across endothelial cells [21,28,29]. The important role of IL-5 in the development of eosinophilic inflammation has been established, as described above. GM-CSF also plays roles in eosinophil activation. For example, GM-CSF induces eosinophil O_2_^−^ generation and degranulation in the presence of VCAM-1 or intercellular cell adhesion molecule 1 in vitro [31]. Furthermore, the airway eosinophils obtained after segmental allergen challenge in allergic subjects can be activated by GM-CSF, but not by IL-5, mainly because IL-5 receptor (R) α is downregulated [39,40].

Moreover, lipid mediators, such as leukotriene (LT) B_4_, cysLTs, and platelet-activating factor, may contribute to the accumulation of tissue eosinophils in the airways. As for cysLTs, LTE_4_ inhalation increases the accumulation of eosinophils [28,29]. LTD_4_ enhances the expression of β2 integrins on human eosinophils in vitro, and thus augments eosinophil adhesion [41]. Moreover, LTD_4_ directly induces transendothelial migration, respiratory burst, and degranulation, mainly via the cysLT1 receptor and β2 integrin [42]. These findings suggest that, in addition to the type 2 network, lipid mediators, including cysLTs, can induce eosinophilic infiltration and activation in the airways.

## 4. Role of CCR3 Ligands or CCR5 Ligands in the Eosinophil Accumulation in EP

The constitutive and/or inducible expression of CCR1, CCR3, CCR5, CXC chemokine receptor (CXCR) 1, CXCR2, CXCR3, and CXCR4 in human peripheral blood eosinophils has been reported [38,43,44,45]. However, proteomic studies of eosinophils found or described only CCR1, CCR3, and CXCR4 as being present on human blood eosinophils [46,47]. Although the reason for this discrepancy is unknown, it may be due to differences in the methods of receptor expression measurement or eosinophil purification, or the characteristics of the donors, and so on.

Among these receptors, CCR3 is highly and constitutively expressed on eosinophils. CCR3 ligands, such as eotaxin (CCL11), eotaxin-2 (CCL24), eotaxin-3 (CCL26), RANTES (CCL5), MCP-2 (CCL8), MCP-3 (CCL7), and MCP-4 (CCL13), can induce eosinophil recruitment in allergic inflammation [43]. Pretreatment of eosinophils with anti-CCR3 antibody diminishes their chemotactic response to CCR3 ligands in vitro [43]. Cellular sources of CC chemokines are likely epithelial cells, fibroblasts, and mononuclear cells [28,29,33,48]. Furthermore, Katoh et al. reported that the concentration of eotaxin was elevated in the BALF of EP patients [35]. Tateno et al. also reported that the concentration of eotaxin was increased in the BALF of CEP patients [36]. Therefore, CCR3 ligands, including eotaxin, may contribute to the pathogenesis of EP (Figure 2).

We found that BALF obtained from AEP or CEP patients induced the transmigration of eosinophils across endothelial cells [37]. This transmigration was blocked by anti-β_2_ integrin mAb. The concentrations of eotaxin-2 and MCP-4, which are CCR3 ligands, were increased in the BALF of EP patients, and anti-CCR3 mAb or anti-MCP-4 mAb suppressed the transmigration of eosinophils induced by the BALF of EP patients. Eotaxin-2 is a potent chemoattractant that binds with high affinity to CCR3 on eosinophils. The expression of eotaxin-2 is increased in airway epithelial cells from asthmatics, and is highest in cases of severe asthma [49]. MCP-4, which is mainly produced by epithelial cells and lymphocytes, is also a potent chemoattractant for eosinophils, and it induces a calcium flux in CCR3-positive cells [50]. Therefore, eotaxin-2 and MCP-4 may be important, specifically for the induction of eosinophil transendothelial migration in EP.

In addition to CCR3, the role of CCR5 in the development of eosinophilic inflammation has been highlighted (Figure 2) [38]. Similar to MIP-1α (CCL3) and MIP-1β (CCL4), RANTES, a CCR3 ligand, also functions as an eosinophil chemoattractant via CCR5 [51], although its expression was not confirmed in proteomic studies [46,47]. MIP-1β differs from the other chemokines in that it is a specific CCR5 ligand [52]. Kobayashi et al. reported that the concentration of the CCR5 ligand MIP-1β is elevated in the BALF of EP patients [38]. Eosinophils constitutively secrete MIP-1β, which induces eosinophil migration, and MIP-1β secretion is enhanced by IL-5 [38]. CCR5-deficient mice show decreased airway eosinophilic inflammation and hyperresponsiveness [53].

## 5. Role of CXCR3 Ligands or CCR4 Ligands in the Development of EP

Other chemokines may also be involved in the eosinophil accumulation in EP. For example, CXCR3 is expressed on eosinophils [45], although its expression was not confirmed in proteomic studies [46,47]. Jinquan et al. and our group have reported that CXCR3 ligands upregulate the effector functions of eosinophils, such as adhesion, O_2_^−^ generation, degranulation, and cytokine/chemokine production [45,54]. Even if CXCR3 is not expressed on eosinophils, there is a possibility that CXCR3 ligands can activate eosinophils via other mechanisms, such as another yet-to-be-defined receptor. As for its role in EP, Katoh et al. reported that CXCR3 ligands, such as interferon (IFN) γ-inducible protein of 10 kDa (IP-10; also known as CXCL10) and monokine induced by IFN-γ (Mig; also known as CXCL9), and CXCR3-expressing eosinophils were increased in the BALF of CEP patients [55]. However, treatment with an anti-CXCR3 mAb did not inhibit the transendothelial migration of eosinophils induced by the BALF of EP patients [37], suggesting that CXCR3 ligands play a negligible role in the eosinophil transendothelial migration in EP.

The concentrations of CCR4 ligands, such as thymus and activation-regulated chemokine (TARC; also known as CCL17), are increased in the BALF and serum of EP patients [15,56]. Furthermore, increased TARC levels are detected in the peripheral circulation of AEP patients, even when blood eosinophilia is absent [57], suggesting that an increase in TARC would precede eosinophilic infiltration. CCR4 is preferentially expressed on Th2 cells, but not on eosinophils, and thus it is assumed that CCR4 ligands could induce eosinophilic inflammation through the activation of Th2 cells. A CCR4 antagonist did not modify the eosinophil transendothelial migration induced by the BALF of EP patients [37]. This finding suggests that, although CCR4 ligands are upregulated in the BALF of EP patients, they do not directly induce eosinophil accumulation in EP.

Interestingly, several reports have suggested that the expression of chemokine receptors on eosinophils differed between eosinophils from peripheral blood and those from airways [55,58,59]. For example, Liu et al. reported that airway eosinophils demonstrated increased CXCR3 or CCR4 expression and decreased CCR3 expression when compared to eosinophils from peripheral blood [58]. There were higher percentages of CXCR3-expressing eosinophils in BALF than in peripheral blood [55]. These findings strongly indicate that airway eosinophils express more CXCR3 or CCR4 and less CCR3 when compared to peripheral blood eosinophils. Therefore, especially in airways, CXCR3 ligands or CCR4 ligands may play roles in the migration and activation of eosinophils. Furthermore, some reports have suggested that CCR4 ligands can induce the migration of eosinophils in a CCR4-independent manner, likely through another yet-to-be-defined receptor [60].

## 6. Role of Type 2 Cytokines (IL-4 and IL-13) in the Development of EP

IL-4 and IL-13 contribute to eosinophil accumulation in the airway; however, the accumulation is thought to be induced by an indirect mechanism, as it is generally accepted that IL-4 and IL-13 with physiological concentration do not activate eosinophils directly. We also confirmed that IL-4 and IL-13 do not activate eosinophil functions, such as adhesion or O_2_^−^ generation, in vitro in eosinophils obtained from the blood of healthy volunteers (unpublished data). In contrast, IL-4 and IL-13 induced VCAM-1 expression in endothelial cells and the production of CCR3 ligands from epithelial cells, fibroblasts, and mononuclear cells (Figure 2), as described above. The levels of IL-13 and, in some reports, IL-4 are increased in the BALF of EP patients [14,15,17,19]. Therefore, IL-4 and IL-13 play roles in the accumulation of eosinophils in the lung, not by directly activating eosinophils, but by inducing VCAM-1 expression in endothelial cells and CCR3 ligand expression.

In eosinophil-dominant severe asthma, anti-IL-4Rα mAb, which blocks both IL-4 and IL-13 signaling, strongly reduced acute exacerbations and increased lung function [61]. Furthermore, in severe chronic rhinosinusitis with nasal polyps, anti-IL-4Rα mAb reduced polyp sizes, sinus opacification, and the severity of symptoms [62]. Considering the role of IL-4/IL-13 in the development of eosinophilic airway inflammation, anti-IL-4/IL-13 treatment likely suppresses the accumulation of eosinophils in tissues in vivo, although it has not yet been established.

## 7. Role of Eosinophil Growth Factors/Cytokines in the Development of EP

IL-5, IL-3, and GM-CSF contribute to eosinophil accumulation via their survival-enhancing effects on eosinophils [21]. The final accumulation of eosinophils in the airway is a function of both eosinophil recruitment to the airway and their continued survival there. Furthermore, eosinophil growth factors including IL-5 play roles in the priming of eosinophils before transmigration. IL-5 caused the shape of blood eosinophils to change in vitro [63]. Moreover, IL-5 upregulated the production of reactive oxygen species that are induced by eotaxin (CCL11) or RANTES (CCL5) [64], suggesting that IL-5 may prime eosinophils for a greater response to chemoattractants (CCR3 ligands). The level of IL-5, but not GM-CSF, is increased in the BALF of EP patients [12,13,14,15,16,17,19,37]. However, anti-IL-5 mAb did not suppress the eosinophil transendothelial migration induced by the BALF of EP patients [37], suggesting the lack of a role for IL-5 in the chemotactic movement in EP. Rather, IL-5 likely contributes mainly by priming eosinophils and enhancing their survival in lung parenchyma (Figure 2).

## 8. Role of Lipid Mediators in the Development of EP

Lipid mediators may play roles in the development of EP. LTB_4_, LTD_4_, and platelet-activating factor are potent chemoattractants for eosinophils [28,29], and the expression of both LTB_4_ and cysLTs is increased in EP [65]. We found that LTB_4_ was upregulated in the BALF of EP patients, and the transmigration of eosinophils induced by BALF was suppressed by an LTB_4_ receptor antagonist [37]. Therefore, LTB_4_ may contribute to the enhanced eosinophil transmigration induced by the BALF of EP patients (Figure 2). Montelukast (leukotriene receptor antagonists; LTRA; also known as cysLT receptor antagonists) did not modify the transmigration of eosinophils induced by the BALF of EP patients [37], suggesting that cysLTs play a negligible role in the eosinophil transendothelial migration in EP.

## 9. Possible Role of Damage-Associated Molecular Pattern Molecules in the Development of EP

As inhaled agents are considered to work in a non-specific (non-allergic) manner, an innate immune response, rather than an adaptive immune response, may be involved in the pathogenesis of EP.

Stressed or damaged cells release damage-associated molecular pattern molecules (DAMPs), which function as endogenous danger signals that alert the innate immune system to unscheduled cell death and microbial invasion [66]. Several reports have suggested a role for DAMPs in the activation of eosinophils [67,68,69]. Damaged epithelial cells can induce eosinophilic migration, degranulation, and cytokine production, likely through the release of DAMPs from stressed or damaged cells [67]. We reported that uric acid (UA) and adenosine triphosphate (ATP), which are important DAMPs, activated eosinophil functions [68,69]. IL-33 is released from damaged cells in response to stress conditions, such as infection, injury, and inflammation [70], and directly activates eosinophil functions [71].

In the BALF of EP patients, the concentration of UA is increased, and it correlates with the number of eosinophils and IL-5 concentrations [18]. Furthermore, the concentration of ATP is increased in the BALF of EP patients, and it correlates with the UA concentrations [18]. The concentration of IL-33 is also increased in EP [17,18], and it correlates with the UA and ATP concentrations [18].

## 10. Possible Role of Extracellular Matrix Protein in the Development of EP

Periostin is an extracellular matrix protein that is highly expressed in the airways of asthmatics in response to type 2 cytokines, including IL-13 [72]. It has been investigated as a biomarker of type 2-mediated asthma [72,73,74]. It functions as a matricellular protein [72] that binds to cellular receptors and activates cells, including eosinophils. Johansson et al. and our group have reported that periostin induced eosinophil adhesion, O_2_^−^ generation, and degranulation through the αMβ2 integrin [75,76]. Furthermore, IL-5-activated eosinophils migrate on periostin in an ADAM8-dependent manner [77], suggesting that periostin has a role in the eosinophil accumulation in EP. Actually, the concentration of periostin is elevated in the BALF of EP patients [19,78], and it correlates with the eosinophil and lymphocyte counts and the concentrations of IL-5, IL-13, and transforming growth factor β1 [19].

Recent studies have suggested that the amphiregulin-osteopontin axis may play roles in the development of eosinophilic airway inflammation [79]. Amphiregulin-producing pathogenic memory Th2 cells can induce airway fibrosis through osteopontin, an extracellular matrix protein produced by inflammatory eosinophils [79]. In fact, while eosinophils produce osteopontin [80], osteopontin induces eosinophil accumulation through the α4 integrin in vitro [81]. The concentration of osteopontin is also elevated in the BALF of EP patients [82].

## 11. Conclusions

Eosinophils must adhere to and migrate across vascular endothelial cells to accumulate in airways. VCAM-1 is an adhesion molecule expressed on endothelial cells, and it is specifically important for eosinophil adhesion. CCR3 ligands, including eotaxin-2 (CCL26) and MCP-4 (CCL13), and CCR5 ligands, including MIP-1β (CCL4), all may play important roles in eosinophil migration. IL-4 and IL-13 are important for eosinophil accumulation, because they increase the expression of VCAM-1 and the expression of CCR3 ligands. IL-5 contributes to eosinophil accumulation by priming eosinophils and enhancing their survival. LTB_4_, DAMPs, including UA and ATP, and extracellular matrix proteins, including periostin, have been found to, or suggested to, play roles in the accumulation or activation of eosinophils in EP. Although anti-IL-5 treatment and anti-IL-4/IL-13 treatment are clinically available for treating eosinophil-dominant severe asthma, recent research has been mainly focused on molecules that are more upstream, such as thymic stromal lymphopoietin. Therefore, further investigations are needed to clarify the whole picture, including these upstream molecules, to fully understand the mechanism of eosinophilic accumulation in EP.

## Figures and Tables

**Figure 1 biomolecules-10-00638-f001:**
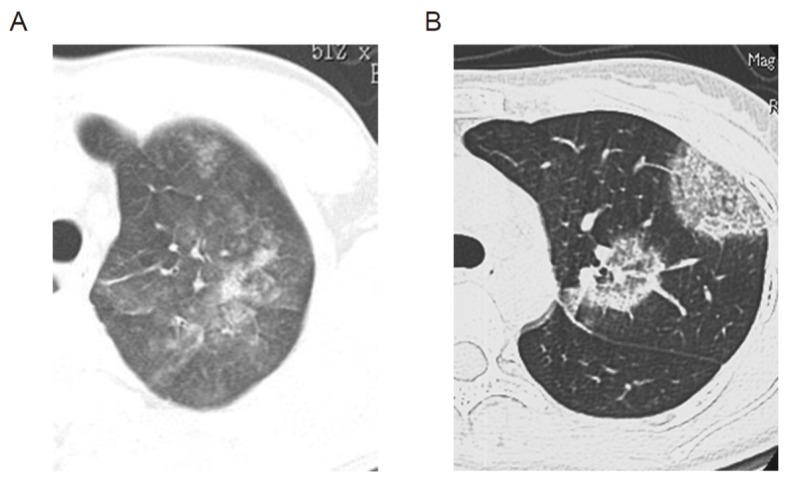
Findings of computed tomography (CT) in acute eosinophilic pneumonia (AEP) and chronic eosinophilic pneumonia (CEP). (**A**) shows the findings of CT in AEP. Ground-glass opacity/airspace consolidation and interlobular septal thickening are representative CT findings of AEP. (**B**) shows the findings of CT in CEP. Bilateral or unilateral airspace consolidation predominantly in the peripheral region (photographic negative pattern) is a representative CT finding of CEP.

**Figure 2 biomolecules-10-00638-f002:**
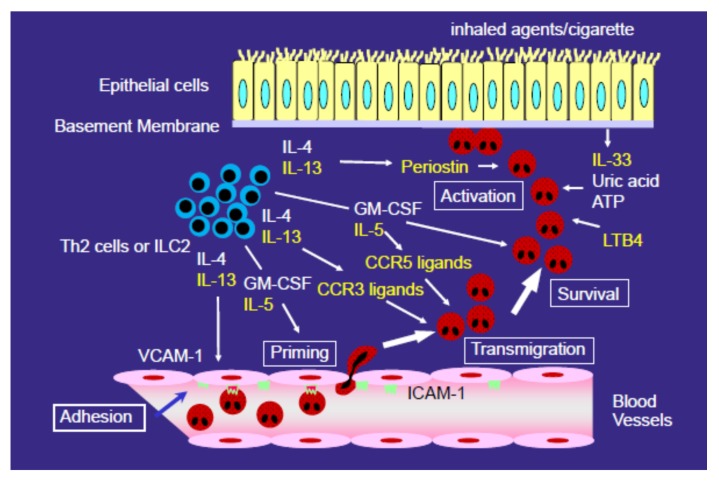
Possible mechanisms of eosinophil accumulation in the airway in eosinophilic pneumonia. Eosinophils must adhere to and migrate across vascular endothelial cells to accumulate in airways. vascular cell adhesion molecule (VCAM)-1 is an adhesion molecule expressed on endothelial cells, and it is specifically important for eosinophil adhesion. Cc chemokine receptor (CCR)3 ligands, including eotaxin-2 (CCL26) and MCP-4 (CCL13), and CCR5 ligands, including MIP-1β (CCL4), all play important roles in eosinophil migration. IL-4 and IL-13 are important for eosinophil accumulation, because they increase the expression of VCAM-1 and the expression of CCR3 ligands. IL-5 contributes to eosinophil accumulation by priming eosinophils and enhancing their survival. LTB_4_, damage-associated molecular pattern molecules (DAMPs), including IL-33, and extracellular matrix proteins, including periostin, have been found or suggested to play roles in the accumulation or activation of eosinophils. Periostin directly activates eosinophils and induces migration, and thereby plays a role in the accumulation of eosinophils in the airway. Yellow font indicates what we consider to be important mediators/molecules in the development of EP.

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
