# Peer review of "Possible Mechanisms of Eosinophil Accumulation in Eosinophilic Pneumonia"

_biomolecules, 2020, doi:10.3390/biom10040638_

Round 1

Reviewer 1 Report

The paper is a comprehensive review of existing literature on the role of eosinophils in eosinophilic pneumonia. It is well written, although some paragraphs contain irrelevant details and other are too brief. Below there are issues that should be addressed or clarified.

  1. The review lacks the aim, could the Authors add it to the paper?
  2. Taking into account that Biomolecules comprise readers of wide range of molecular background and of different specialties not restricted to lung diseases, the review should contain a brief introduction about EP – its frequency, classification, symptoms and etiology, so the reader could assess whether it is common or rare clinical problem; whether it is homogenous with defined background or comprises heterogeneous diseases.
  3. The paragraph about IL-4 and IL-13 role in eosinophils accumulation in the airways seem too abbreviated – particularly taking into account the figure where IL-4/IL-13 seem to play a central role in eosinophil accumulation (transmigration, activation, adhesion). Moreover, some sentences are unclear and require completion: “However, it is generally believed that IL-4 or IL-13 have negligible roles in the activation of eosinophil functions in vitro. Therefore, IL-4 and IL-13 play roles in the accumulation of eosinophils in the lung mainly via an indirect mechanism” – what mechanism? Author should comment on that. As previously shown by studies in allergic asthma these cytokines play a direct role in eosinophil accumulation in the airways – could the authors add a paragraph on Th2 cytokines/response in the context of eosinophilic inflammation in pneumonia?
  4. The References are overabundant – it seems that one-third of cited papers is out-of-date (papers older than 20 yrs), so they may not be crucial for recent state-of-the art in EP. Possibly the Authors could check the reference list and revise it taking into account the large number (over 90 cited papers), so the Authors could decide which may be irrelevant (or out of date) and remove them from the paper.
  5. The conclusion is a list of analyzed chemokines and proteins involved in the pathogenesis of EP, but it should rather contain a brief summary of the reviewed literature as well as suggest some hints for future directions in eosinophilic pneumonia research. Could the Authors revise it?

Minor points:

Figure 1 – it is helpful to summarize the text, but the legend should be more detailed e.g. the different color of fonts for different cytokines/chemokines

English language requires revision: too much sentences in the passive voice, long, with the verb at the end – it is difficult to read

Reviewer 2 Report

General comments:

This review manuscript reviews literature, and describes and suggests possible mechanisms for eosinophil accumulation in the lung in eosinophilic pneumonia (EP), which includes the possible or likely participation of chemokines and their receptors, cytokines, lipid mediators, danger signals, and extracellular matrix (ECM) proteins.

Specific comments:

Major comments:

  1. Figure 1: As the manuscript and the figure discuss possible mechanisms of accumulation in the airway, not just movement of eosinophils to it, it would be an advantage to add the concept of Survival (e.g., in a box, like Adhesion, Transmigration, and Activation) to the figure, and point out and draw arrow(s) from possible or suggested survival-promoting factors (e.g., IL5, as mentioned in the text). One would assume that the final accumulation in the airway is a function of both recruitment there and continued survival once the eosinophils are there.
  2. Fig. 1, contd.: Also, it seems reasonable to think that activation happens or at least starts, so at least priming happens, before transmigration. The eosinophil shown as having changed shape and beginning transmigration presumably has already been activated or at least primed. Of relevance, e.g., IL5 causes shape change of blood eosinophils (Am. J. Respir. Cell Mol. Biol. 2014, 50:654-64), Thus, I suggest moving the box Activation to before, i.e., below Transmigration (and if so, possibly have a box Continued or Prolonged Activation after Transmigration), or insert a new box Priming before Transmigration.
  3. Fig. 1, additional: It could be illustrated more clearly in the figure that periostin possibly or suggestedly promotes migration and accumulation of eosinophils in the lung tissue (see also below).

Minor comments:

4. Please also give the formal names for the chemokines (e.g., CCL11 for eotaxin-1)

5. Sections 1 and 2: Please give some numbers as to the prevalence in the population of all EP, acute and chronic EP (AEP and CEP).

6. Introduction: Molds are stated as being possible inducers of EP. Is it known if fungal PAMPs (pathogen-associated molecular patterns), e.g., cell wall components such as beta-1,3-glucans, can induce EP?

7. Page 2, line 49: Regarding the corticosteroid therapy, is this inhaled, oral, and/or injected?

8. Lines 72-74: Were the case reports on compassionate use of anti-IL5 and which antibody, mepolizumab? Have there been any trials of any phase (I-III) with anti-IL5 or anti-IL5 receptor in EP?

9. P. 3, line 100: Please expand the text somewhat here to indicate that the lack of activation of airway eosinophils by IL5 is due to downregulated IL5 receptor alpha, as described in the given references (40-41).

10. A recent review (Larose M-C et al. 2017, Front. Med. 4:136) and the proteomic study on eosinophils (J. Proteome Res. 2016, 15:1524-33) describes or finds only CCR1, CCR3 and CXCR4 as present on human blood eosinophils. Please discuss this in the context of your manuscript (e.g., lines 109-111) and your references.

11. Line 136 says eosinophils secrete MIP-1beta (CCL4), but again CCL4 was not detected in the proteomic study. Please comment.

12. Line 149: A typo, should be “thymus” (not “tymus”). Please proofread the text of any other possible errors.

13. Line 180: What do you mean by “manifestation of the effector functions”? You may mean that IL5 can prime eosinophils for greater response to chemoattractants. If so, please state clearly (and give reference(s)).

14. Lines 185-187: The text here is potentially confusing and superficially contradictory. First it is stated that an LTB4 receptor antagonist suppressed the activity by BALF, then that leukotriene receptor antagonists did not modify it. In the second instance, you may mean specifically cysLT receptor(s). Please clarify.

15. Line 213: AlphaMbeta2-mediated adhesion of eosinophils to periostin was first reported in Am. J. Respir. Cell Mol. Biol. 2013, 4:503-10. Later articles demonstrated that periostin also supports migration of cytokine-activated eosinophils, of potential and likely importance for eosinophil recruitment to the airway, and does so in a manner involving the metalloproteinase ADAM8 (Clin. Exp. Allergy 2017, 47:1263-74 and PLoS One 2018, 13:e0201320).

16. Line 221: What is/are the likely osteopontin receptor(s) on eosinophils?

17. Line 226: I suggest writing “have been found or suggested to play roles…” (not just “found”), since most of the described mechanisms are possible suggested ones based on various in vivo and in vitro data but cannot necessarily be said to be “found” to play definite roles in EP.

Reviewer 3 Report

This submission as review article provides a comprehensive review of different promising pathways related to eosinophil accumulation in eosinophilic pneumonia. As seen, the authors are very familiar with research literature in this area of investigation and therefore they use many references (92) concerning older and current papers.

This is a clear, concise, well-conceived and well-written manuscript appropriately illustrated and accurately referred.

I have only one minor comment

  • Page 1, line28 – I suggest a revision of keywords and to complete suitable ones (e.g. pneumonia is missing)

Author Response

Comments:

This submission as review article provides a comprehensive review of different promising pathways related to eosinophil accumulation in eosinophilic pneumonia. As seen, the authors are very familiar with research literature in this area of investigation and therefore they use many references (92) concerning older and current papers.

This is a clear, concise, well-conceived and well-written manuscript appropriately illustrated and accurately referred.

I have only one minor comment

Page 1, line28 – I suggest a revision of keywords and to complete suitable ones (e.g. pneumonia is missing)

Author response

We agree with the reviewer’s comments. According to the suggestion, we added: “eosinophilic pneumonia” and “pneumonia” to the keywords in the revised manuscript.

Round 2

Reviewer 2 Report

General comment:

This revised manuscript has addressed all previous comments.